# Prevalence and Characteristics of Psoriasis in Romania—First Study in Overall Population

**DOI:** 10.3390/jpm11060523

**Published:** 2021-06-07

**Authors:** Alin Codruț Nicolescu, Ștefana Bucur, Călin Giurcăneanu, Laura Gheucă-Solovăstru, Traian Constantin, Florentina Furtunescu, Ioan Ancuța, Maria Magdalena Constantin

**Affiliations:** 1Roma Medical Center for Diagnosis and Treatment, 011773 Bucharest, Romania; nicolescualin66@yahoo.com; 2III^rd^ Department, “Carol Davila” University of Medicine and Pharmacy, 050474 Bucharest, Romania; calin.giurcaneanu@gmail.com (C.G.); traianc29@yahoo.com (T.C.); florentina.furtunescu@umfcd.ro (F.F.); iancuta@hotmail.com (I.A.); drmagdadinu@yahoo.com (M.M.C.); 3II^nd^ Department of Dermatology, Colentina Clinical Hospital, 020125 Bucharest, Romania; 4Department of Dermatology, Elias University Emergency Hospital, 011461 Bucharest, Romania; 5Department of Dermatology, “Sf. Spiridon” Clinical Hospital, “Grigore T. Popa” University of Medicine and Pharmacy, 700115 Iași, Romania; lsolovastru13@yahoo.com; 6Department of Urology, “Th. Burghele” Hospital, 050652 Bucharest, Romania; 7Department of Public Health and Management, 050463 Bucharest, Romania; 8Department of Rheumatology, “Dr. I. Cantacuzino” Clinical Hospital, 020475 Bucharest, Romania

**Keywords:** psoriasis vulgaris, prevalence, comorbidities, risk factors

## Abstract

*Background:* Psoriasis is a chronic inflammatory disease characterized by an excessive hyperproliferation of keratinocytes and a combination of genetic, epigenetic, and environmental influences. The pathogenesis of psoriasis is complex and the exact mechanism remains elusive. *Objectives:* The study of the prevalence of psoriasis will allow the estimation of the number of people suffering from this condition at the national level, as well as the development and validation of a questionnaire to estimate the prevalence and the risk factors associated with the disease. *Methods:* A quantitative research was conducted at a national level among the target population in order to validate the questionnaire and estimate the national prevalence. *Results:* Declaratively, the prevalence of psoriasis in the studied group (N = 1500) is 4%, the first symptoms appearing around the age of 50, with a certified diagnosis being made on average at 55 years. The prevalence of psoriasis vulgaris was 4.99%. *Conclusions:* The results obtained will be useful in guiding future initiatives and communication campaigns related to this condition, and the methodological approach used will provide the opportunity to make recommendations for improving similar initiatives in the future.

## 1. Introduction

Psoriasis is a chronic, inflammatory, genetically determined skin condition, with a frequency of 1–3% in the general population [1]. The onset of the disease frequently occurs around the age of 20 or around the age of 40, and the negative consequences of psoriasis should be emphasized: in young people at the beginning of their careers it often leads to their underestimation, and in working adults it often leads to premature retirement. Among the dermatological conditions, psoriasis takes the first place in terms of deterioration of life quality index, ahead of malignant skin diseases. Psoriasis impairment to psychological quality of life is comparable to cancer, myocardial infarction, and depression [2]. Psoriasis therefore has a debilitating influence both physically and mentally on the patient’s daily life. Standard diagnostic criteria for epidemiological studies of psoriasis are currently lacking. In their absence, clinical examination and diagnosis of psoriasis by dermatologists provides the gold standard to underpin epidemiological research in psoriasis [3].

Psoriasis has been declared by the World Health Organization the fifth most important chronic non-contagious disease, along with diseases such as diabetes, cancer, cardiovascular or respiratory diseases. Statistics show that there are over 125 million patients in the world and Romania ranks first in Europe, having approximately 400,000 people with this condition [4]. A total of 20–30% of patients with psoriasis also develop psoriatic arthritis, which over time leads to severe, deforming joint injuries, which often cause disability [5]. In the absence of a set of specific criteria for the diagnosis of psoriatic arthritis, the most commonly used method for recognizing and monitoring this condition remains the clinical aspect [6].

In Romania, psoriasis remains a common, immune-mediated disease, with increasing prevalence. This publication quantifies the prevalence of psoriasis in Romania, its potentially genetic background, and the importance of environmental factors in triggering it [7], including the role of the diet [8,9] and other factors, such as vitamin D deficiency [10].

Psoriatic subjects represent an important target for health care issues and further epidemiological studies. The estimates provided can help guide countries and the international community when making public health decisions on the appropriate management of psoriasis.

## 2. Objectives

It is thought that there are more undiagnosed or untreated patients beyond those who have been diagnosed. This study aims to develop and validate a screening questionnaire for the early, presumptive identification of psoriasis in the general population and to estimate the prevalence of people suffering from this condition at a national level.

In order to achieve the objectives, quantitative research was conducted at a national level among the general adult population (people over 18 years old in urban and rural areas in Romania).

## 3. Materials and Methods

The study was performed in two stages. Stage 1 involved the development of the questionnaire and construction of the predictive model. Within this stage, the questionnaire was developed by a mixed team of experts (dermatologists, epidemiologists, sociologists), the methodology was finalized, and standardized working procedures were developed.

Stage 2 involved the validation of the questionnaire and the estimation of the national prevalence. At this stage, a national study was conducted on a representative sample of 1500 adult individuals. From this total of 1500 people to whom the questionnaire was applied, a subgroup of 500 people was randomly selected and received in addition a clinical examination aiming to diagnose the eventual presence of psoriasis.


Detailed description of the study.


The first stage of the study was conducted between November 2018–February 2019, on two groups of subjects: the group with psoriasis (288 patients with psoriasis selected from the offices of dermatologists from all over the country) and the control group (222 healthy people randomly selected from the general population). The inclusion criteria were: adults over 18 years old, from rural and urban areas in Romania, without a diagnosis of psoriasis (healthy group); or those with a previous diagnosis of psoriasis (affected group). The only exclusion criterion was represented by institutionalized persons. Both groups received a questionnaire applied through face-to-face interviews and underwent parallel dermatologic examination.

The questionnaire included three sections: (i) a general part referring to demographic characteristics; (ii) a screening part with specific questions regarding the existence of symptoms and the history of dermatological diseases; (iii) a section of a detailed characterization of the person from a social and clinical point of view. During this first stage, both the research tool (questionnaire) and the logistical organization and field activity management tools were tested. This activity was carried out by field operators with experience in conducting household surveys.

A predictive model has been elaborated by performing a logistic regression applied to the pilot data. The discriminative variables were: (1) the symptoms of psoriasis (presence and nature of skin and nail lesions, including the occurrence of stress for all forms of psoriasis) and the symptoms and signs of arthritis for the arthropathic form; (2) the presence of hereditary collateral antecedents; (3) smoking at the time of the study; (4) history of pharyngitis in childhood for the presence of arthropathy; (5) history of obesity, but not obesity at the time of the study; (6) depression due to the arthropathic nature of psoriasis; (7) treatment; (8) some psycho-emotional and contextual characteristics. The proposed questionnaire was reviewed and updated according to the relevant variables. 

The second stage of the study was conducted in 2019 on a group of 1500 respondents over the age of 18 on a nationally representative group. The national study used a three-stage stratified probabilistic grouping model. The questionnaire was applied in a “face-to-face” interview by specialized interview operators. A total of 500 randomly selected individuals from this group underwent clinical examination performed by a dermatologist at the nearest medical unit to the home of the person included in the group. The application of the questionnaire at the same time with the performance of clinical examinations allowed the validation of the questionnaire and the establishment of its specific parameters (sensitivity, specificity, accuracy, predictive value).

In a further step, the prevalence of psoriasis was estimated at the national level. 

The national sample selection used a stratified three-step probabilistic grouping pattern in which places were selected with PPS (probability proportional to size) in the first step. Households were systematically selected in the second step and only one individual from each household was selected by SRS (simple random sample) using mobile data collection devices. The first step consisted in the selection of localities (stratification was performed according to the environment of residence (urban or rural) and type of urban/rural locality: in urban areas, stratification was performed by size of locality: small town (less than 50,000 inhabitants), medium town (50,000–199,999 inhabitants) and large city (over 200,000 inhabitants). In rural areas, the stratification was carried out taking into account two types of rural localities: localities with up to 5000 inhabitants and localities with over 5000 inhabitants, depending on the number of questionnaires required, the sampling points were calculated for each locality. In the following step, starting from the selected group points, the systematic selection of households that were included in the research was performed. Prior to this stage, the households were identified and included in a list from which the selection of those that were part of the group was made (mapping and listing). In the last step, 1500 individuals were randomly selected. 

The interviews were conducted with people over the age of 18 who actually lived at the address selected from the group. In each identified household, the questionnaire was applied to one person. If there were several eligible persons in the household (more than two over the age of 18) the questionnaire was applied to one person who was randomly selected. If the selected person was not at home at the time of the first visit and was absent for less than 30 days (the person was at work, on a delegation, on holiday etc.), a new visit to the household was scheduled. In this regard, at least three attempts (visits) were made to interview the selected person. The validation of the group was performed according to the official statistical data provided by the National Institute of Statistics [11].

Within the project, a software application was developed that allowed both data collection and management of the activity of conducting clinical examinations at a national level. The application was a client server type. The samples and the collected data were stored in the server component both after applying the questionnaire and as a result of clinical examinations. The clients of the application were represented by the face-to-face interview operators and by the doctors involved in performing the clinical examinations.

The coordination, monitoring, and supervision of the field activity were carried out by a field coordinator with experience. The activity which involved data collection at a local level was coordinated and supervised by the county coordinators, the data collection being ensured by a specialized network of interview operators. Dermatologists and nurses were recruited to perform clinical examinations from the dermatology medical offices who expressed their acceptance to participate in the study. After performing the clinical examinations, the specialists sent the results to the headquarters according to the established working procedures. 

The statistical analysis was performed with the R system. Mean ± standard deviation and/or median have been calculated for the scale variables and proportions for the qualitative ones. Comparisons were performed using the *t*-test and ANOVA for quantitative research, normally distributed variables and the chi-square or Fisher test for qualitative. The logistic regression has been used to construct the predictive model. Sensitivity and specificity have been calculated for the questionnaire in relation to the gold standard dermatological diagnosis for psoriasis. A 95% confidence interval was calculated for the psoriasis prevalence in general population.

The demographic profile included: gender, age group, level of education, occupation, residence environment (urban, rural), region of economic development (Northeast, Southeast, South, Southwest, West, Northwest, Center, Bucharest). The clinical profile and personal medical history of the respondents included details on symptoms, frequency of symptoms, determining factors, associated conditions, previous diagnosis of dermatological disease, the age of onset of symptoms, personal medical history. The use of medical services for dermatological diseases was analyzed in terms of the following characteristics: previous treatments for dermatological conditions and their type, specific clinical examinations performed, accessed services.

The diagnosis in psoriasis for the subjects performing the dermatological examination was a clinical one (observation of erythematous and squamous patches, generalized or localized in specific areas, itching, burning or soreness, thickened, pitted or ridged nails, swollen and stiff joints). During the medical consultation, personal and family history and risk factors were also evaluated. A subgroup of 500 people randomly selected were invited to participate in a specialized clinical examination performed by a dermatologist in a medical office in order to analyze the association between the questionnaire and the real status of the subjects and to explore the validity and prediction of the questionnaire. The subgroup had similar geographic and demographic characteristics and similar history of the disease (Table 1).

Selected subjects were asked to give their consent for the examination. From the 500 selected subjects, 18 did not agree to participate to the clinical examination and another 21 agreed, but did not show up at the dermatologist. The study algorithm is shown in Figure 1. 

The diagnosis in psoriasis was a clinical one (observation of erythematous and squamous patches, generalized or localized in specific areas, itching, burning or soreness, thickened, pitted or ridged nails, swollen and stiff joints). During the medical consultation personal and family history and risk factors were also evaluated.

## 4. Results

Declaratively, the prevalence of psoriasis in the studied group (N = 1500) reached 4.2%, the first symptoms appearing around the age of 50 and a competent diagnosis being required by the patients around the age of 55. A series of studies carried out all over the world suggests that Caucasians are more affected than other races, with higher prevalence percentages, unlike Australian Aborigines, the pre-Colombian population of the New World, Andean Indians, Amerindians, Alaskan, Canadian, or Native Americans of the United States, where psoriasis has been reported to be extremely rare or absent [12]. Interestingly, late-onset psoriasis is slightly more common than the early-onset type, contrary to the often quoted convention that 75% of new psoriasis cases are present before the age of 40 [13].

Neither the environment, urban/rural, nor the type of locality presents statistical data or is associated with the presence of psoriasis for the 1500 individuals or for the 461 remaining subjects from the subgroup of 500 randomly selected individuals in the second stage of the study (Table 2). In contrast with an Italian study [14], psoriasis did not seem to be homogeneously distributed across the northern, central, and southern geographical areas of the country, findings which are similar to a study in Spain [15]. These variations in prevalence rates between regions are not unusual and are explained by different genetic and/or financial and stress conditions. 

The psoriatic patients, but not the healthy group, were usually supervised by the doctor, with a few exceptions (although they were selected by a doctor, perhaps the exceptions indicate a lack of understanding of the question). With rare exceptions, psoriatic patients report that they follow a treatment; no subject who is considered healthy is considered to be under treatment. Patients with psoriasis had significantly more frequent relatives with psoriasis than the healthy group. 

Skin lesions appeared significantly more frequently in psoriatic patients than healthy people, but also in 3/4 of healthy people. Lesions of the elbows or knees have been reported very frequently by psoriatic patients and very rarely by healthy people. To a lesser extent, the lesions on the scalp, hands, feet, back, and to a very small extent other skin lesions had a differentiating value. An association with psoriasis in general is also observed for scales, pruritus, red areas, bleeding areas, frequent in psoriasis, but quite common in healthy people. The same association shows the disappearance with local treatment. Persistence of lesions does not appear to be associated with psoriasis, but the occurrence of lesions under stress, including pruritus, was reported more frequently by psoriatic patients. Increasing evidence over the past decade has shown that pruritus can be one of the most prevalent and burdensome symptoms associated with psoriasis, affecting almost every patient to some degree [16]. Spontaneous disappearance is relatively rare, but more common in healthy people. Small nail stains are reported significantly more frequently by psoriatic patients than healthy ones and nail loss is rare in both psoriatic and healthy individuals (Table 3).

Comorbidities and strong related conditions are detailed in Table 4. Smoking is significantly more common in psoriatic patients than healthy people and also, the age of the onset of smoking does not make a difference. There is no difference between a former smoker or a passive smoker and the gross variables regarding the intensity of smoking exposure do not show, individually, differences between the studied groups. The link between alcohol consumption seems vague, the only more convincing association being with the concern of loved ones in the last year, which is significantly more common in those with psoriasis. The report of pharyngitis in childhood was insignificantly more frequent in psoriasis, but the reports of hospitalizations for pharyngitis in childhood were significantly more common in psoriasis. There were no significant differences in height, body weight, or maximum body mass in the groups. The diagnosis of depression as well as the treatment were significantly more frequently associated with psoriasis. Patients with mild psoriasis can experience psychiatric comorbidities; however, depression is more common in patients with severe psoriasis or psoriatic arthritis [17].

Of the 500 subjects selected for clinical examination, 461 attended the visit.

In this study group, the prevalence for psoriasis vulgaris was 4.99% (2.95: 7.03). The concordance between the initial diagnosis, established on the basis of the questionnaire and the final clinical diagnosis was 99.13% for psoriasis vulgaris. An acceptable sensitivity of the questionnaire more than 85% and a specificity of almost 100% were identified.

Regarding the treatment for psoriasis, only 2 of the clinically confirmed cases of the disease reported following treatment, while 7 did not receive treatment and 15 did not answer the question. Only one subject who considered himself healthy declared to be under treatment. Some psoriasis patients, even some dermatologists, are quite reluctant to undertake biologic therapy [18], even though it has demonstrated real efficacy in treatment of psoriasis and psoriatic arthritis [19], but novel biologics act by novel targets, technology, and mechanisms of action compared to previously approved biologics and the explosive development of biological therapy and the emergence of biosimilars, revolutionary tools against the most serious and provocative diseases which represent a significant success in the effort to provide advanced healthcare to patients all over the world [20,21,22]. To complete these results, an interesting study provides useful data on widely used biologic drugs and their tolerability, discontinuation rate, and the incurrence of severe adverse events [23].

We have detailed the data obtained from the national study regarding healthy and non-healthy groups in Table 5. Regarding the hereditary factors, patients with psoriasis significantly more frequently had relatives with psoriasis compared to those without disease. This result was also confirmed in the initial group. The frequency of reported skin and nail lesions differ significantly between the psoriatic patients and the disease-free, unlike the initial group in which the yellowing of the nails was significantly associated with the certified diagnosis of psoriasis. Regarding smoking, the only significant difference was identified for smoking exposure in the last 12 months, present in 21.1% of psoriatic patients and in none of the healthy patients. Moreover, alcohol consumption in the last year has never been significantly associated with an increased likelihood of the disease. Regarding infections, the data correspond to the results of stage 1. Many systemic therapies available for the management of psoriasis patients who cannot be treated with more conservative options, such as topical agents and/or phototherapy, can worsen or reactivate a chronic infection. Therefore, before administering immunosuppressive therapies it is mandatory to screen patients for some infections, including hepatitis B or C [24,25]. Metabolic syndrome is a highly prevalent, multifaceted condition characterized by a constellation of abnormalities that include abdominal obesity, hypertension, dyslipidemia, and elevated blood glucose [26]. The frequency of obesity was not significantly different between the psoriatic and the non-psoriatic patients (Table 5). Psychological stress has long been shown to play an important role in the natural history of psoriasis, but the details of this relationship remain to be clearly defined [27]. In the initial group, 5% of the interviewees stated that they often felt that they could not cope with the important things in life. In the diagnostic group, the proportion of those who chose the “very often” option on this question was 4.2% in non-patients and 8.7% in those with psoriasis, respectively. The difference did not reach statistical significance (*p* = 0.342). 

## 5. Discussion

In our study we elaborated a screening questionnaire for the early presumptive diagnosis of psoriasis and we applied this tool on a randomly selected sample of 1500 subjects from the general population. 

In a second stage of the study, a group of 500 subjects (comparable to the initial sample by gender, age, area of residence, region, level of education) was selected to be examined clinically at the same time, for a certain diagnosis. Among the analyzed risk factors, a significantly higher presence of family history in patients was also confirmed, in accordance with the predictive model, while for the rest of the analyzed factors there were no significant differences between psoriatic and non-psoriatic patients.

Our study has some limitations related to the fact that only one third of the initial sample has received a medical examination due to feasibility reasons. We tried to minimize those limits by randomly selecting the subjects and therefore revealing the similarities of the two groups from demographic and geographical perspectives. Another limitation is related to the fact that only 23 patients with psoriasis have been identified in the subgroup of people with a medical examination. We used for comparisons in this case the Fisher test and the demographic and geographical characteristics did not differ significantly for psoriatic and non-psoriatic patients. Last but not least, we did not measure the concordance between different dermatologists who have been involved in the clinical diagnosis. 

## 6. Conclusions

The pilot study demonstrated the ability of the questionnaire and the procedure for completing it to highlight the most expected associations based on the literature on the epidemiology of psoriasis.

Our study is the first attempt, upon our knowledge, to estimate the prevalence of psoriasis in Romania. The prevalence of psoriasis vulgaris measured within the second stage of the study was 4.99% and the proposed questionnaire was found to have a convenient sensitivity for psoriasis vulgaris (86.96%). The correlation between the declared diagnosis established on the basis of a questionnaire and the final clinical diagnosis was 99.13% for psoriasis vulgaris.

Further research is required to determine the reasons driving the increase in psoriasis prevalence over time. The results obtained in this study are intended to be a source of recommendations and suggestions for new initiatives, campaigns, and public policy proposals to address the various issues of these diseases.

## Figures and Tables

**Figure 1 jpm-11-00523-f001:**
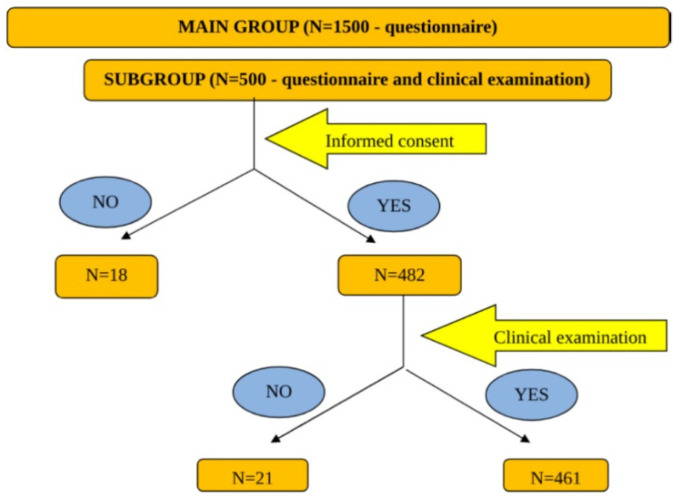
The study algorithm.

**Table 1 jpm-11-00523-t001:** Comparative analysis of the groups according to the main demographic variables in stage 2 of the study.

Criteria	No.	%	No.	%	*p*-Value *
Region					
Bucharest-Ilfov	173	11.5%	58	11.6%	
Center	176	11.7%	59	11.8%	
Northeast	261	17.4%	87	17.4%	
Northwest	191	12.7%	63	12.6%	
South	220	14.7%	73	14.6%	
Southeast	193	12.9%	64	12.8%	
Southwest	146	9.7%	50	10.0%	
West	140	9.3%	46	9.2%	
Total	1500	100.0%	500	100.0%	>0.99
Environment					
Rural	638	42.5%	210	42.0%	
Urban	862	57.5%	290	58.0%	
Total	1500	100.0%	500	100.0%	0.834
Gender					
Men	726	48.4%	241	48.2%	
Women	774	51.6%	259	51.8%	
Total	1500	100.0%	500	100.0%	0.938
Age groups					
18–29	264	17.6%	92	18.4%	
30–49	611	40.7%	202	40.4%	
50+	625	41.7%	206	41.2%	
Total	1500	100.0%	500	100.0%	0.921
Education					
No studies	2	0.1%	1	0.2%	
Primary	102	6.8%	35	7.0%	
Secondary	650	43.3%	222	44.4%	
Tertiary	746	49.7%	242	48.4%	
Total	1500	100.0%	500	100.0%	0.947
Declared diagnostic history					
Yes	63	4.2%	21	4.2%	
No	1437	95.8%	479	95.8%	
Total	1500	100.0%	500	100.0%	>0.99

***** Test Chi^2^.

**Table 2 jpm-11-00523-t002:** Demographic characteristics depending on the presence of psoriasis vulgaris.

Variable	No.	%	*p*-Value–Nonhealthy-Healthy
Region			
Bucharest-Ilfov	56	12%	0.089
Center	40	9%	
Northeast	87	19%	
Northwest	55	12%	
South	69	15%	
Southeast	59	13%	
Southwest	49	11%	
West	46	10%	
Total	461	100%	
Environment			
Rural	194	42%	0.251
Urban	267	58%	
Total	461	100%	
Gender			
Men	221	48%	0.104
Women	240	52%	
Total	461	100%	
Age groups			
18–29	81	18%	0.188
30–49	189	41%	
50+	191	41%	
Total	461	100%	
Education			
No studies	1	0%	0.76
Primary	34	7%	
Secondary	201	44%	
Tertiary	225	49%	
Total	461	100%	

**Table 3 jpm-11-00523-t003:** Results and data analysis of the national study (N = 1500 respondents).

Monitored by a Doctor	Local Treatment	Family Member with Psoriasis	Skin Lesions 25%		Types of Lesions 26%	Characteristics of the Lesions	Nail Changes
47.9%	44%	A form of psoriasis9%	Hands 48%	Elbows 19%	Erythema 82%	Disappear spontaneously 52%	Nail yellowing 19%
75.8%— by a dermatologist	Only on lesions 88%	Psoriatic arthritis 2%	Feet 46%	Knees 17%	Pruritus 69%	Disappear and reappear depending on the local treatment 52%	Nail stains 19%
15.2%—by a family doctor	Phototherapy 17%		Face 30%	Nails 15%	Scales 33%	In correlation with stress 39%	Onychomadesis/nail loss 6%
6.1%—other doctors	NSAIDs 13%		Back 25%	Genitals 13%	Pigmentation disorders 29%	Persistent without improvement 36%	
	Other 4%		Abdomen 24%	Ears 9%	Papules/Pustules18%		
			Thorax 20%	All over the body 6%	Hemorrhagic areas 17%		
			Scalp 19%	Other areas 6%	Other 8%		

**Table 4 jpm-11-00523-t004:** Comorbidities and related conditions resulting from the national study (N = 1500 respondents).

Comorbidities and Related Conditions					
**Arterial hypertension**	31%	Following a treatment 82%			
**Dyslipidemia**	22%	Following a treatment 42%			
**Diabetes mellitus**	10%	Following a treatment 85%			
**Depression/Anxiety**	10%	Following a treatment 45%			
**Infections**	30% pharyngitis and tonsillitis during childhood	12% hospitalized for these diseases			
**Obesity**	17% yes, 3% from childhood	1 out of 5 was diagnosed with obesity at some point in life			
**Stress, life style**	1 out of 3 cannot cope with the important things in life	2 out of 5 sometimes feel that the difficulties have accumulated to such an extent that they can no longer control them			
**Alcohol**	Never 23%	1 time/month 36%	2–4 times/month 29%	>4 times /month 12%	
**Smoking**	Average age of onset—20 years 45%	Currently smokers 51%	Average smoking period—15 years	Passive smoking 25%	Exposure to smoke during childhood 74%

**Table 5 jpm-11-00523-t005:** Results and data analysis of the national study (N = 1500 respondents).

	Nonhealthy		Healthy		*p*-Value
	Number	%	Number	%	
**Hereditary factors**					
Relative with psoriasis	17	70.8%	32	7.3%	<0.001
**Skin and nail lesions**					
Skin lesions	8	33.3%	118	27.0%	0.54
Nail lesions	4	16.7%	79	18.1%	0.58
Nail yellowing	4	16.7%	88	20.1%	0.46
Onychomadesis	2	8.3%	23	5.3%	0.38
**Smoking**					
Ever smoked for at least 2 years	11	45.8%	165	37.8%	0.279
Currently smoker	3	12.5%	82	18.8%	0.248
Former smoker	8	33.3%	80	18.3%	0.179
Exposure in the last 12 months	0	0.0%	92	21.1%	0.004
Exposure in childhood	18	75.0%	325	74.4%	0.581
**Alcohol**					
Alcohol consumption ever in the last year	16	66.7%	332	76.0%	0.303
**Infections**					
Pharyngitis/tonsillitis during childhood	11	45.8%	102	23.3%	0.028
Hospitalization	6	25.0%	57	13.0%	0.009
**Obesity**					
BMI > 25	18	75.0%	224	51.3%	0.216
Ever been obese	8	33.3%	74	16.9%	0.053
Obesity during childhood	0	0.0%	18	4.1%	0.375
**Arterial hypertension**	9	37.5%	146	33.4%	0.663
**Diabetes mellitus**	4	16.7%	45	10.3%	0.307
**Depression/Anxiety**	5	20.8%	40	9.2%	0.073
**Dyslipidemia**	10	41.7%	115	26.3%	0.104

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
