# Peer review of "Prevalence and Characteristics of Psoriasis in Romania—First Study in Overall Population"

_jpm, 2021, doi:10.3390/jpm11060523_

Round 1
Reviewer 1 Report
An interesting study of prevalence in the Romanian population;
The paper seems to have various methodological problems that needs to be assessed. I suggest the authors carefully recheck all the data reported;
I have some queries:
I found the study methodology very hard to understand and quite complex; I suggest the authors improve this part, better explaining and simplifying the various parts.
For example , stage 2 has 500 or 1500 participants as declared in the table?
Plus I think table 5 is wrong ; if I understood correctly, in stage 2 of the 461 patients included, onli 4,99%, so I guess 23, were affected by psoriasis.
You said that healthy patients are basically 23 and the others are non-healthy ....I think you switched groups.
A revision of the paper done by an English native speaker would in my opinion improve a lot the readability of the paper.
Results should be section 4 instead of 3...please check.
Page 7 line 243-247; a small paragraph better highlighting possible biologic drugs in use for psoriasis should be added; here some articles you could consider: doi: 10.1371/journal.pone.0241575. doi: 10.1111/dth.14504. and https://doi.org/10.3390/healthcare9050543
Thank You
Reviewer 2 Report
This study aims to investigate the prevalence of psoriasis and the estimation of the number of people suffering from this condition in Romania, as well as the development and validation of a questionnaire to estimate the prevalence and the risk factors associated with the psoriasis. The authors however did not provide a detailed scientific background to support the aim of their study. Both Methods and statistical analysis should be explained in more detail. Moreover, it is known that there is a nutritional influence on psoriasis that should be addressed (even in limitations). In addition, the effect on the clinical severity of psoriasis is also the result of other factors not evaluated by the authors, such as waist circumference and body composition. The lack of all this basic information must be addressed by the authors. English needs to be improved.
Here are my suggestions:
- Introduction: Introduction does not provide sufficient background information to allow the reader to understand and evaluate results of this study without needing to refer to previous publications on the topic. In particular, psoriasis is known as a multifactorial disease where genetic and environmental factors play a part. Smoking and nutritional status are well-established environmental risk factors for psoriasis. The authors should add a paragraph on the importance of main environmental risk factors in psoriasis (see and cite: Environmental Risk Factors in Psoriasis: The Point of View of the Nutritionist. Int J Environ Res Public Health, PMID: 27455297), including the role of diet and/or single nutritional factors, such as vitamin D (see and cite:
- Vitamin D and its role in psoriasis: An overview of the dermatologist and nutritionist. Rev Endocr Metab Disord. 2017, PMID: 28176237;
- Very low-calorie ketogenic diet (VLCKD) in patients with psoriasis and obesity: an update for dermatologists and nutritionists. Crit Rev Food Sci Nutr. 2020, PMID: 32969257
- Coffee consumption, metabolic syndrome and clinical severity of psoriasis: good or bad stuff? Arch Toxicol. 2018, PMID: 29594327).
- METHODS
- How was psoriasis diagnosed?
- Have psoriasis medications been evaluated? How?
- Were treatment naïve patients present?
- The protocol number of local Ethics Committee is missing.
- What inclusion/exclusion criteria were evaluated?
- The validation of the groups was performed according to the official statistical data provided by the National Institute of Statistics. Which? Please indicate the reference.
- Within the project, a software application was developed that allowed both data collection and management of the activity of conducting clinical examinations at the national level. Please specify better what it is.
- Was the study conducted without sponsorship?
- How were obesity, alcohol and smoking assessed?
- Smoking is the most important variable in the study, how was it assessed? according to ...? Questionnaire? Which? Specific references are missing
- Has physical activity been assessed? How?
Furthermore: A flow chart could be appropriate to demonstrate how participants were included in the study.
- Statistical analysis must be detailed. Please specify all statistical tests used for which analysis.
- Results not have been clearly presented.
- Table 1: What do the percentages in the healthy subjects column indicate?
- Table 4. How were comorbidities assessed? Were the subjects taking medications for these conditions?
- Diabetes mellitus: Which? type 1, type 2?
- Obesity: what degree?
- Tables and Figures are lacking a description of statistical analyses in their footnote.
- Discussion needs re-organizing and authors should provide potential explanation for each results.
- Limitations to the study must be added as well as any strengths.
- English language needs improvement. I would recommend to advice a native English speaker.
Round 2
Reviewer 1 Report
The article improved with this new revision. The article needs still to be formatted according to journal style. After editing, the article in my opinion may be published.
Reviewer 2 Report
I have no further comments